# “Growth-Promoting Effect” of Antibiotic Use Could Explain the Global Obesity Pandemic: A European Survey

**DOI:** 10.3390/antibiotics11101321

**Published:** 2022-09-28

**Authors:** Gábor Ternák, Márton Németh, Martin Rozanovic, Gergely Márovics, Lajos Bogár

**Affiliations:** 1Institute of Migration Health, Medical School, University of Pécs, H-7624 Pécs, Hungary; 2Department of Anesthesiology and Intensive Care, Medical School, University of Pécs, H-7624 Pécs, Hungary; 3Department of Public Health Medicine, Medical School, University of Pécs, H-7624 Pécs, Hungary

**Keywords:** obesity, childhood obesity, adult obesity, microbiome, dysbiosis, antibiotics, antibiotic consumption, growth-promoting effect, obesity pandemic

## Abstract

Clinical observations indicated a higher rate of obesity among children who received antibiotics at early ages. Experimental studies supported the role of the modified gut microbiome in the development of obesity as well. For identifying antibiotic classes that might promote or inhibit obesity-related dysbiosis, a database of the average yearly antibiotic consumption (2008–2018) has been developed using the European Center for Disease Prevention and Control (ECDC) yearly reports of antibiotic consumption in the community for the major antibiotic classes in 30 European countries, which were compared to the childhood and adult obesity prevalence featured in the Obesity Atlas. Pearson’s chi-square test was applied to estimate positive/negative correlations between antibiotic consumption and obesity. One-way ANOVA has been applied to test the differences in antibiotic consumption between groups, and logistic regression analysis was performed to determine the odds ratios (OR) of antibiotic consumption for obesity. Strong, positive associations were estimated between childhood obesity and the total consumption of systemic antibiotics, broad-spectrum, beta-lactamase-resistant penicillin, cephalosporin, and quinolone, and a negative correlation was found with the consumption of tetracycline, broad-spectrum, beta-lactamase-sensitive penicillin, and narrow-spectrum, beta-lactamase-sensitive penicillin. Our observation indicated that the “growth-promoting effect” of the consumption of certain antibiotic classes might be identified as a possible etiology in the development of obesity and might be the explanation for the obesity “pandemic”.

## 1. Introduction

The human race has struggled to overcome food scarcity, disease, and hostile environments for centuries, but after the industrial revolution and the improving living conditions with the availability of nutrients, this situation changed. The recognition of obesity as a disease was theoretically established in 1948 by the World Health Organization (WHO), but for several years, medical professionals had ignored its importance [1]. The so-called Fogarty report (1973) in the USA [2] initiated further investigations of the emerging obesity epidemics. Although obesity did not attract the attention of the mass media until recent decades, its prevalence in industrialized countries began to increase progressively early in the last century. Clear scientific evidence emerged, as the alarming trend in obesity rates was provided by the regular, national surveys performed since 1960 [3]. Over the past ~50 years, the prevalence of obesity has increased worldwide to pandemic levels [4]. According to WHO reports (2021), worldwide obesity has nearly tripled since 1975; 39% of adults aged 18 years or over were overweight in 2016, and 13% were obese. A total of 39 million children under the age of 5 were overweight or obese in 2020 (available at https://www.who.int/news-room/fact-sheets/detail/obesity-and-overweight, accessed on 22 March 2022). Body mass index (BMI) is a simple weight-for-height index commonly used to classify overweightness and obesity (kg/m^2^). According to the WHO definition, overweightness is considered when the BMI is greater than or equal to 25, and obesity is a BMI greater than or equal to 30. For children aged between 5–19 years, overweightness is considered when the BMI-for-age is greater than one standard deviation above the WHO Growth Reference median, and obesity is estimated when the BMI is greater than two standard deviations above the WHO Growth Reference median (available at https://www.who.int/news-room/fact-sheets/detail/obesity-and-overweight, accessed on 22 March 2022).

Obesity is a complex health issue resulting from a combination of causes and individual factors such as behavior and genetics. The most common causes of overweightness and obesity are well-known: genetics, the lack of physical activity, dietary patterns (particularly the sales of those foods currently implicated in the pandemic’s origins and spread, such as sugar-sweetened beverages and fast foods with high caloric density, including French fries), medication use (steroids), certain diseases (hypothyreosis), etc. [5].

The accidental discovery of the growth-promoting effect of antibiotics [6] soon followed (1950) with the mixture of antibiotics to animal fodder to achieve higher and faster weight gain in food animals [7], which quickly resurfaced in the environment, polluting potable water sources and food [8]. It might be strongly suspected that if feeding antibiotics to food animals produce weight gain, humans receiving antibiotics as therapeutics or from the environment might suffer a similar, inadvertent “weight gain”, appearing as an obesity pandemic. One of the authors (Ternak, G.) raised this concept in 2004 [9]. The association between antibiotic consumption and obesity, particularly antibiotic exposure in early childhood, is well documented in several publications [10,11,12,13,14,15].

It is suspected that the modification of the gut microbiome results in “leaky gut syndrome”, which could allow bacteria, toxic digestive metabolites, bacterial toxins, and small molecules to “leak” into the bloodstream and act through the gut-brain axis (GBA), facilitating the development of obesity and several other ailments [16]. Antibiotics are known to disrupt microbiota composition, and while a rapid recovery is observed following short-term antibiotic treatment, pervasive effects may be obtained after repeated antibiotic perturbations [17,18].

As different classes of antibiotics produce different modifications to the microbiome, it might be suspected that obesity-related dysbiosis is attached to certain classes of antibiotics, particularly in childhood obesity [19,20].

## 2. Objectives/Hypothesis

Based on the above associations, we suspected that different classes of antibiotics might promote or inhibit the development of obesity-related dysbiosis, and the dominant consumption of “promoting” or “inhibiting” antibiotics in European countries might influence the prevalence of obesity in children and adults of the given countries. We aimed to identify antibiotic classes that might promote or inhibit obesity-related dysbiosis.

To evaluate the relevance of our hypothesis, we have compared the antibiotic consumption data of major antibiotic classes to adult and childhood obesity prevalence Figure 1 and Figure 2 in European countries.

## 3. Materials and Methods

Standard Protocol Approvals, Registrations, and Patient Consent: Publicly available, already published statistical data on antibiotic consumption (ECDC) and the prevalence of childhood/adult obesity prevalence (Obesity Atlas) had been compared and statistically analyzed, which did not require any ethical approval. The study did not contain any patient data, and no patient consent was requested.

Antibiotic consumption data were extracted and calculated as the average yearly consumption from the ECDC yearly reports of antibiotic consumption in the community for 30 European countries, which regularly reported consumption data to the ECDC, from 2008 to 2018. The total average yearly consumption of antibiotics for systemic use (J01) was expressed in terms of Defined Daily Dose/1000 Inhabitants/ Day (DID), and the relative share of the consumption of major antibiotic classes has been calculated in terms of the percentage of the total consumption at Anatomical Therapeutic Chemical classification (ATC) level three and level four within the group of J01C (penicillin). The major classes included were tetracycline (J01A), penicillin (J01C), broad-spectrum, beta-lactamase-sensitive penicillin (J01CA), broad-spectrum, beta-lactamase-resistant combination penicillin (J01CR), narrow-spectrum, beta-lactamase-sensitive penicillin (J01CE), narrow-spectrum, beta-lactamase-resistant penicillin (J01CF), cephalosporin (J01D), macrolides (J01F), and quinolones (J01M), covering the majority (over 90%) of the antibiotics used in the community. Obesity prevalence data for children 5–9 years of age were extracted from the Atlas of Childhood Obesity 2019 October (available at: http://s3-eu-west-1.amazonaws.com/wof-files/11996_Childhood_Obesity_Atlas_Report_ART_V2.pdf, accessed on 22 March 2022), and the data for adults were extracted from http://gamapserver.who.int/gho/interactive_charts/ncd/risk_factors/obesity/atlas.html (accessed on 22 March 2022).

Statistical analysis: Pearson’s chi-square test was applied to calculate correlations. A significant correlation (positive/negative) was estimated when *p* values were equal to or less than 0.05 (≤0.05), and we considered a non-significant (moderate) correlation when the *p* values remained between 0.05 and 0.09 (0.05 ≤ 0.09).

Countries were divided into three groups based on their rankings of childhood and adult obesity and to perform variance analysis. A one-way ANOVA test was performed to estimate a statistically significant difference between the obesity groups. A significant difference was estimated when *p* values were equivalent to or less than 0.05 (≤0.05).

Childhood obesity data were divided into two groups based on means, creating below-mean and above-mean categories. The same methodology was followed for adult obesity. Logistic regression analysis was performed to determine the odds ratio (OR, CI 95%) for each antibiotic class for both childhood and adult obesity. A significant result was estimated when *p* values were equal to or less than 0.05 (≤0.05).

Spreadsheets containing the antibiotic consumption databases and the obesity prevalence Figure 1 and Figure 2 have been developed, along with the rank order (first 10 positions) of the prevalence of childhood obesity and the rank order of antibiotic consumption. Scatter diagrams have been plotted to show the association between childhood obesity data and antibiotic consumption (Figure 1 and Figure 2).

## 4. Results

Pearson’s chi-square test indicated a strong, positive (“promoter”) correlation between childhood obesity and the total consumption of antibiotics for systemic use in the community (J01, *r:* 0.517, *p*: 0.003), broad-spectrum, beta-lactamase-resistant combination penicillin (J01CR, *r*: 0.573, *p*: 0.001), cephalosporin (J01D, *r*: 0.539, *p*: 0.002), and quinolone (J01M, *r*: 0.554, *p*: 0.001). A negative (“inhibitor”) correlation has been observed between childhood obesity and tetracycline (J01A, *r*: −0.497, *p*: 0.005), broad-spectrum, beta-lactamase-sensitive penicillin (J01CA, *r:* −0.369, *p*: 0.03), and narrow-spectrum, beta-lactamase-sensitive penicillin (J01CE, *r*: 0.373, *p*: 0.042). A moderate, negative association was observed with the narrow-spectrum, beta-lactamase-resistant penicillin (J01CF, *r*: −0.339, *p*: 0.067) (Table 1 and Table 2).

The rank order of countries (Table 3) shows the first ten positions of childhood obesity prevalence, as compared to the rank order of “promoter” and “inhibitor” classes of antibiotic consumption. Concordance (correlation) was observed between the higher prevalence of childhood obesity and the countries with higher consumption of “promoter” antibiotics and lower consumption of “inhibitor” antibiotics.

The groups formed by these ranks have been studied with one-way ANOVA to demonstrate the differences in antibiotic consumption between them. In childhood obesity, we found a statistically significant difference in beta-lactamase-resistant combination penicillin (J01CR, *p*: 0.009) and quinolone (J01M, *p*: 0.003). We have found a statistically significant difference between macrolides (J01F, *p*: 0.048) and adult obesity as well.

The logistic regression analysis showed a statistically significant elevated risk with the consumption of beta-lactamase-resistant combination penicillin (J01CR, OR: 1.175, CI95% 1.044–1.323, *p*: 0.007) and quinolone (J01M, OR: 1.252, CI95% 1.017–1.540, *p*: 0.034). On the other hand, there is a statistically significant decreased risk in the case of tetracycline (J01A, OR: 0.846, CI95% 0.733–0.977, *p*: 0.023). No similar association was found between antibiotic consumption and adult obesity.

## 5. Discussion

The worldwide epidemic of obesity has become an important public health issue, which bears down heavily on national health budgets regarding several, serious complications. Obesity is a multifactorial disorder in which various elements (genetic, host, and environment) play a definite role, but the exact pathology is not appropriately identified. The most frequent explanation for obesity is the consumption of “junk food” and the lack of physical activity, but this does not explain the more-than-two-fold difference in the prevalence of childhood obesity rates in European countries (Italy: 17.7%, Sweden 8.6%). The difference for adults is lower (Malta 28.9%, Denmark 19.7%).

Experiments demonstrated the role of the microbiome in the development of obesity and diabetes. It has been observed that lipopolysaccharide (LPS), which is produced by certain bacteria, is also capable of inducing “metabolic endotoxemia”, and this effect does not occur in the absence of CD14 cells in genetically modified mice [21,22].

The gut microbiome plays an important role in energy homeostasis, regulating energy harvesting, fat deposition, as well as feeding behavior, and appetite. Feeding, appetite, and energy expenditure are under the control of the central nervous system (CNS), which receives various peripheral signals of energy status and its availability, such as gut hormones and adipokines (signaling molecules released by the adipose tissue) [23,24].

Apart from the orexigenic hormones (appetite stimulants), such as ghrelin, neuropeptides, etc., the hypothalamus regulates the homeostatic feeding behavior, while other neural brain regions, such as the insular cortex, orbitofrontal cortex, nucleus accumbens, amygdala, and dopaminergic ventral tegmental area, present groups of neurons implicated in the reward-related, non-homeostatic control of feeding [25].

Food intake is mainly regulated by the energy needed from the brain based on its adenosine triphosphate (ATP) disposability and dysfunctions in any part of this metabolic pathway, such as congenital leptin deficiency, which can result in a persistent state of positive energy balance and obesity development [26,27].

The eubiotics/dysbiosis condition of the gut microbiota strongly influences our health and disease status. As the gut microbiota is a crucial factor in human physiology, many of these effects are mediated by metabolites that are either produced by the microbes or derived from the transformation of environmental or host molecules [28].

Our study identified major classes of antibiotics (J01, J01CR, J01D, J01M), which might contribute to the development of obesity-promoting dysbiosis in childhood obesity and, probably, macrolides (J01F) in adults through different molecular mediators and GBA, which might “promote” the development of obesity. In the case of the total consumption of antibiotics for systemic use, it might be considered that the majority of the classes of antibiotics consumed belong to the “promoter” group (J01CR, J01D, and J01M); this is why, in this category (J01), a positive correlation was observed. At the same time, we have found antibiotic classes that might inhibit the development of obesity-promoting dysbiosis (J01A, J01CA, and J01CE). Antibiotic classes showing negative associations with obesity figures (Figure 2) might inhibit the microbial production of molecules promoting obesity. Our findings are supported by the observation that the higher consumption of “promoter” antibiotics is found in countries (mostly the Mediterranean countries) with a higher prevalence of childhood obesity (Table 3). On the other hand, Scandinavian countries, using mostly penicillin and tetracycline, show less of a prevalence of childhood obesity.

As J01CR (broad-spectrum, beta-lactamase-resistant combination penicillin) and J01CA (broad-spectrum, beta-lactamase-sensitive penicillin) classes of antibiotics promote and inhibit the development of childhood obesity, respectively, the role of added beta-lactamase inhibitors (clavulanic acid, sulbactam) changes the direction of the effect of the antibiotics.

In adult obesity, only ANOVA analysis indicated a positive correlation between the consumption of macrolide antibiotics and adult obesity data (BMI ≥ 30).

Animal studies and meta-analyses of human studies on the association between antibiotics and the subsequent development of obesity are suggestive of a link between exposure to antibiotics—particularly, early exposure in life—and the development of subsequent obesity as a result of alterations in the diversity of gut microbiota. The evidence is strong in animal models, whereas evidence in humans is inconclusive, requiring well-designed, long-term longitudinal studies to examine this association [13]. Experimental studies in animal models as well as in humans have shown that obesity is associated with a decrease in the abundance of Bacteroidetes and an increase in the number of Firmicutes. Based on recent meta-analyses and epidemiologic studies in healthy children, factors such as the administration of antibiotics during the first 6 months of life, repeated exposure to antibiotics for ≥3 courses, treatment with broad-spectrum antibiotics, and male gender have been associated with increased odds of overweightness/obesity [29].

In one of our previous publications [30], we observed similar associations. Our present study raises the issue that humans, particularly children, might be subjected to the same “growth-promoting effect” of certain classes of antibiotics that are widely utilized in animal husbandry, and it might serve as a unified explanation for the obesity “pandemic”.

The restoration of the microbiome in obesity through fecal microbiome transfer (FMT) could be considered a therapeutic option [31], but the results are controversial [32]. A clinical trial with the use of FMT did not show a reduction in obesity in adults [33].

Current evidence suggests that disruptions to the composition and maturation of the gut microbiota caused by antibiotic use in early life are a key mechanism linking the association between antibiotics and obesity [34].

The limitations of our study are that the results could not be supported at the individual level; only statistical correlations and concordance could be established. However, our findings are in accord with the reports featured in several publications.

## 6. Conclusions

Our comparative analyses identified the classes of antibiotics that might promote or inhibit the development of obesity through the alteration of the microbiome. Mediator molecules arising from the dysbiotic microbiome, acting through the gut-brain axis, could influence the prevalence of obesity. This observation might explain the origin of the obesity pandemic.

## Figures and Tables

**Figure 1 antibiotics-11-01321-f001:**
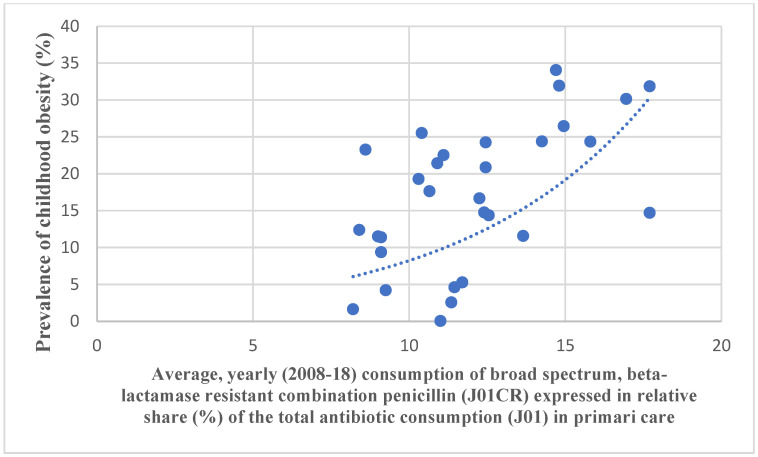
A significant positive correlation was observed between the consumption of broad-spectrum, beta-lactamase-resistant penicillin (J01CR) and the prevalence of childhood obesity.

**Figure 2 antibiotics-11-01321-f002:**
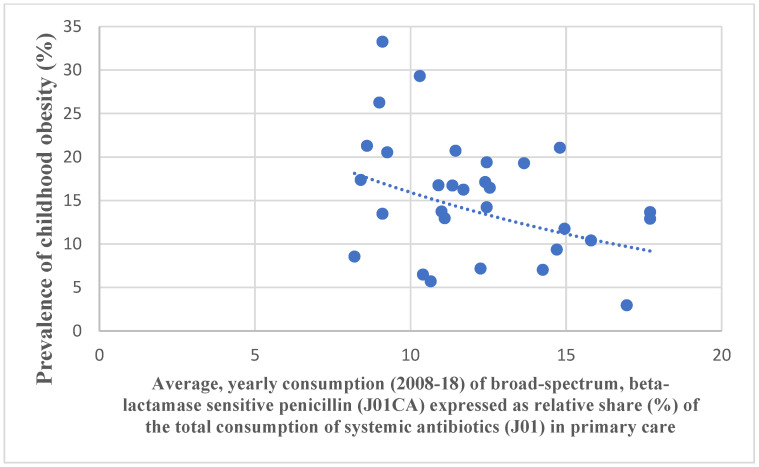
A strong, negative correlation was observed between the consumption of broad-spectrum, beta-lactamase-sensitive penicillin (J01CA) and the prevalence of childhood obesity.

**Table 1 antibiotics-11-01321-t001:** Prevalence of childhood obesity compared to the antibiotic consumption data, and the results of the comparison.

Countries	Childhood Obesity 5–9 Years in % (2019)	Average Yearly Antibiotic Consumption ECDC 2008–2018	Defined Daily Dose/1000 Inhabitants/Day (100%) J01	J01A %	J01C %	J01CA %	J01CR %	J01CE %	J01CF %	J01D %	J01F %	J01M %
**Austria**	10.4	**Austria**	**12.45**	8.19	38.61	6.48	25.52	6.76	0.07	12.95	25.8	0.52
**Belgium**	8.6	**Belgium**	**22.52**	9.17	46.1	21.28	23.27	0.21	1.13	6.64	14.5	10.69
**Bulgaria**	13.65	**Bulgaria**	**17.54**	9.98	31.83	19.28	11.57	1.36	0	17.9	20.2	14.22
**Croatia**	14.95	**Croatia**	**17.86**	6.96	42.57	11.74	26.46	4.32	0.05	17.24	16.7	8.13
**Cyprus**	15.8	**Cyprus**	**26.77**	12.12	34.94	10.39	24.35	0.32	0.09	21.11	11.5	16.77
**Czechia**	12.25	**Czechia**	**16.51**	13.38	35.88	7.16	16.67	11.61	0.23	10.17	22	6.35
**Denmark**	9.25	**Denmark**	**15.24**	10.75	63.09	20.54	4.21	29.46	8.73	0.21	13.2	3.26
**Estonia**	8.4	**Estonia**	**10.23**	16.58	31.73	17.36	12.38	2.05	0.01	10.26	22.8	8.2
**Finland**	11.7	**Finland**	**16.16**	24.72	29.75	16.24	5.27	7.87	0.22	13.54	7.55	4.93
**France**	10.3	**France**	**23.62**	13.69	50.54	29.3	19.3	0.67	1.17	9.27	14.7	7.32
**Germany**	11.35	**Germany**	**13.23**	17.73	25.15	16.72	2.56	5.87	0.08	20.56	18	10.03
**Greece**	17.7	**Greece**	**32.22**	7.42	29.26	13.66	14.7	0.87	0	24.53	26.1	8.12
**Hungary**	14.25	**Hungary**	**13.65**	8.95	33.95	7.03	24.39	2.61	0	14.27	21.7	15.83
**Iceland**	12.55	**Iceland**	**18.76**	20.3	47.87	16.45	14.36	11.66	5.56	2.88	8.42	4.18
**Ireland**	12.45	**Ireland**	**19.51**	14.2	47.34	14.22	20.87	6.24	6.73	6.28	21	4.52
**Italy**	17.7	**Italy**	**22.23**	2.41	45.4	12.9	31.85	0	0.04	10.68	20.8	14.67
**Latvia**	9	**Latvia**	**10.75**	21.22	38.21	26.26	11.51	0.64	0.01	5.02	14.5	9.27
**Lithuania**	9.1	**Lithuania**	**14.97**	10.75	45.78	33.26	9.37	3.27	0.04	9.67	13.3	7.05
**Luxembourg**	11.1	**Luxembourg**	**23.35**	7.89	37.14	12.95	22.53	0.18	0.89	14.9	17.5	11.41
**Malta**	16.95	**Malta**	**19.18**	6.56	33.08	2.94	30.15	0.38	0.24	23.48	20.5	23.14
**Netherlands**	9.1	**Netherlands**	**9.62**	24.48	32.42	13.47	11.38	2.82	4.36	0.39	14.9	8.34
**Norway**	11	**Norway**	**15.28**	19.33	40.04	13.73	0.04	22.09	4	0.64	9.98	3.08
**Poland**	12.4	**Poland**	**20.39**	11.52	33.08	17.11	14.76	1.07	0.03	13.85	20.4	6.4
**Portugal**	14.7	**Portugal**	**17.79**	4.73	46.7	9.35	34.06	0.1	3.24	9.35	17.3	12.45
**Romania**	10.9	**Romania**	**26.14**	3.79	43.56	16.74	21.42	2.86	2.41	17.57	10.6	12.1
**Slovakia**	10.65	**Slovakia**	**20.33**	7.77	30.5	5.7	17.64	6.67	0	21.82	26.8	9.76
**Slovenia**	12.45	**Slovenia**	**11.68**	3	59.92	19.39	24.26	15.04	1.26	2.99	16.5	9.46
**Spain**	14.8	**Spain**	**18.72**	4.85	54.78	21.07	31.94	0.45	1.09	9.47	12.2	13.49
**Sweden**	8.2	**Sweden**	**12.74**	22.5	49.89	8.54	1.63	27.86	11.67	1.27	4.71	5.61
**UK**	11.45	**UK**	**16.95**	26.82	38.41	20.72	4.61	4.76	8.16	2.28	17	2.73
**Pearson’s, *r***			**0.517**	**−0.497**	**0.141**	**−0.369**	**0.573**	** *−0.375* **	**−0.339**	**0.536**	**0.3**	** *0.554* **
**Pearson’s, *p***			** 0.003 **	** *0.005* **	**0.488**	** *0.03* **	** 0.001 **	** *0.042* **	**0.067**	** 0.002 **	**0.109**	** * 0.001 * **
**ANOVA, *p***			**0.084**	**0.840**	**0.680**	**0.186**	** 0.009 **	**0.188**	**0.459**	**0.069**	**0.841**	** 0.003 **
**OR**			**1.128**	** *0.846* **	**1.013**	**0.898**	** 1.175 **	**0.932**	**0.853**	**1.084**	**1.106**	** 1.252 **
**CI 95%**			**0.965–1.319**	**0.733–0.977**	**0.937–1.096**	**0.792–1.019**	** 1.044–1.323 **	**0.833–1.043**	**0.652–1.118**	**0.973–1.208**	**0.959–1.275**	** 1.017–1.540 **
**OR, *p***			**0.131**	**0.023**	**0.738**	**0.096**	** 0.007 **	**0.221**	**0.249**	**0.142**	**0.166**	** 0.034 **

A significant, positive correlation (bold, framed) was found between childhood obesity and the total consumption of systemic antibiotics (J01), broad-spectrum, beta-lactamase-resistant penicillin (J01CR), cephalosporin (J01D), and quinolone (J01M). A negative correlation (bold, italics, framed) was found between childhood obesity and tetracycline (J01A), broad-spectrum, beta-lactamase-sensitive penicillin (J01CA), and narrow-spectrum, beta-lactamase-sensitive penicillin (J01CE). A moderate, negative correlation was found with narrow-spectrum, beta-lactamase-resistant penicillin (J01CF, bold, italics). ANOVA analysis and OR indicated a positive significance between childhood obesity and the consumption of broad-spectrum, beta-lactamase-resistant, combination penicillin and quinolone (J01CR, J01M). A negative significance was estimated with tetracycline (J01A) (underlined). Glossary: J01: antibiotics for systemic use, J01A: tetracycline, J01C: penicillin, J01CA: broad-spectrum, beta-lactamase-sensitive penicillin, J01CR: broad-spectrum, beta-lactamase-resistant penicillin, J01CE: narrow-spectrum, beta-lactamase-sensitive penicillin, J01CF: narrow-spectrum, beta-lactamase-resistant penicillin, J01D: cephalosporin, J01F: macrolides, J01M: quinolones.

**Table 2 antibiotics-11-01321-t002:** Prevalence of adult obesity compared to the antibiotic consumption data, and the results of the comparison.

Countries	Adult BMI ≥ 30, 2022	Average Yearly Antibiotic Consumption ECDC 2008–2018	Defined Daily Dose/1000 Inhabitants/Day (100%) J01	J01A %	J01C %	J01CA %	J01CR %	J01CE %	J01CF %	J01D %	J01F %	J01M %
**Austria**	20.1	**Austria**	**12.45**	8.19	38.61	6.48	25.52	6.76	0.07	12.95	25.8	0.52
**Belgium**	22.1	**Belgium**	**22.52**	9.17	46.1	21.28	23.27	0.21	1.13	6.64	14.5	10.69
**Bulgaria**	25	**Bulgaria**	**17.54**	9.98	31.83	19.28	11.57	1.36	0	17.9	20.2	14.22
**Croatia**	24.4	**Croatia**	**17.86**	6.96	42.57	11.74	26.46	4.32	0.05	17.24	16.7	8.13
**Cyprus**	21.8	**Cyprus**	**26.77**	12.12	34.94	10.39	24.35	0.32	0.09	21.11	11.5	16.77
**Czechia**	26	**Czech Rep.**	**16.51**	13.38	35.88	7.16	16.67	11.61	0.23	10.17	22	6.35
**Denmark**	19.7	**Denmark**	**15.24**	10.75	63.09	20.54	4.21	29.46	8.73	0.21	13.2	3.26
**Estonia**	21.2	**Estonia**	**10.23**	16.58	31.73	17.36	12.38	2.05	0.01	10.26	22.8	8.2
**Finland**	22.2	**Finland**	**16.16**	24.72	29.75	16.24	5.27	7.87	0.22	13.54	7.55	4.93
**France**	21.6	**France**	**23.62**	13.69	50.54	29.3	19.3	0.67	1.17	9.27	14.7	7.32
**Germany**	22.3	**Germany**	**13.23**	17.73	25.15	16.72	2.56	5.87	0.08	20.56	18	10.03
**Greece**	24.9	**Greece**	**32.22**	7.42	29.26	13.66	14.7	0.87	0	24.53	26.1	8.12
**Hungary**	26.4	**Hungary**	**13.65**	8.95	33.95	7.03	24.39	2.61	0	14.27	21.7	15.83
**Iceland**	21.9	**Iceland**	**18.76**	20.3	47.87	16.45	14.36	11.66	5.56	2.88	8.42	4.18
**Ireland**	25.3	**Ireland**	**19.51**	14.2	47.34	14.22	20.87	6.24	6.73	6.28	21	4.52
**Italy**	19.9	**Italy**	**22.23**	2.41	45.4	12.9	31.85	0	0.04	10.68	20.8	14.67
**Latvia**	23.6	**Latvia**	**10.75**	21.22	38.21	26.26	11.51	0.64	0.01	5.02	14.5	9.27
**Lithuania**	26.3	**Lithuania**	**14.97**	10.75	45.78	33.26	9.37	3.27	0.04	9.67	13.3	7.05
**Luxembourg**	22.6	**Luxembourg**	**23.35**	7.89	37.14	12.95	22.53	0.18	0.89	14.9	17.5	11.41
**Malta**	28.9	**Malta**	**19.18**	6.56	33.08	2.94	30.15	0.38	0.24	23.48	20.5	23.14
**Netherlands**	20.4	**Netherlands**	**9.62**	24.48	32.42	13.47	11.38	2.82	4.36	0.39	14.9	8.34
**Norway**	23.1	**Norway**	**15.28**	19.33	40.04	13.73	0.04	22.09	4	0.64	9.98	3.08
**Poland**	23.1	**Poland**	**20.39**	11.52	33.08	17.11	14.76	1.07	0.03	13.85	20.4	6.4
**Portugal**	20.8	**Portugal**	**17.79**	4.73	46.7	9.35	34.06	0.1	3.24	9.35	17.3	12.45
**Romania**	22.5	**Romania**	**26.14**	3.79	43.56	16.74	21.42	2.86	2.41	17.57	10.6	12.1
**Slovakia**	20.5	**Slovakia**	**20.33**	7.77	30.5	5.7	17.64	6.67	0	21.82	26.8	9.76
**Slovenia**	20.2	**Slovenia**	**11.68**	3	59.92	19.39	24.26	15.04	1.26	2.99	16.5	9.46
**Spain**	23.8	**Spain**	**18.72**	4.85	54.78	21.07	31.94	0.45	1.09	9.47	12.2	13.49
**Sweden**	20.6	**Sweden**	**12.74**	22.5	49.89	8.54	1.63	27.86	11.67	1.27	4.71	5.61
**UK**	27.8	**UK**	**16.95**	26.82	38.41	20.72	4.61	4.76	8.16	2.28	17	2.73
**Pearson’s, *r***			**0.122**	**0.056**	**−0.96**	**0.011**	**0.004**	**−0.291**	**−0.132**	**0.268**	**0.183**	**0.246**
**Pearson’s, *p***			**0.52**	**0.767**	**0.113**	**0.956**	**0.902**	**0.118**	**0.488**	**0.153**	**0.334**	**0.191**
**ANOVA, *p***			**0.246**	**0.426**	**0.193**	**0.732**	**0.471**	**0.291**	**0.476**	**0.222**	** 0.048 **	**0.558**
**OR**			**1.005**	**1.002**	**0.962**	**1.023**	**0.992**	**0.957**	**0.913**	**1.030**	**1.090**	**1.027**
**CI 95%**			**0.875–1.155**	**0.902–1.114**	**0.885–1.045**	**0.921–1.138**	**0.920–1.070**	**0.863–1.061**	**0.709–1.174**	**0.931–1.140**	**0.948–1.254**	**0.883–1.193**
**OR, *p***			**0.938**	**0.964**	**0.962**	**0.668**	**0.838**	**0.402**	**0.476**	**0.564**	**0.227**	**0.733**

ANOVA analysis shows a correlation (bold, underlined) between adult obesity and the consumption of macrolide antibiotics (J01F). Glossary: J01: antibiotics for systemic use, J01A: tetracycline, J01C: penicillin, J01CA: broad-spectrum, beta-lactamase-sensitive penicillin, J01CR: broad-spectrum, beta-lactamase-resistant penicillin, J01CE: narrow-spectrum, beta-lactamase-sensitive penicillin, J01CF: narrow-spectrum, beta-lactamase-resistant penicillin, J01D: cephalosporin, J01F: macrolides, J01M: quinolones.

**Table 3 antibiotics-11-01321-t003:** The rank order of the prevalence of childhood obesity (first 10 positions, bold, framed) compared to the rank orders of the obesity “promoter” (J01CR, J01D, J01M) and “inhibitor” (J01A, J01CA) antibiotics.

Countries	Childhood Obesity 5–9 Years in % (2019)	Countries	J01CR %		J01D %		J01M %		J01A %		J01CA %
**Italy**	**17.7**	**Spain**	**31.94**	**Greece**	**24.53**	**Malta**	**23.14**	UK	26.82	Lithuania	33.26
**Greece**	**17.7**	**Portugal**	**34.06**	**Malta**	**23.48**	**Cyprus**	**16.77**	Finland	24.72	France	29.3
**Malta**	**16.95**	**Italy**	**31.85**	* **Slovakia** *	* **21.82** *	**Hungary**	**15.83**	Netherlands	24.48	Latvia	26.26
**Cyprus**	**15.8**	**Malta**	**30.15**	**Cyprus**	**21.11**	**Italy**	**14.67**	Sweden	22.5	Belgium	21.28
**Croatia**	**14.95**	**Croatia**	**26.46**	Germany	20.56	**Bulgaria**	**14.22**	Latvia	21.22	**Spain**	**21.07**
**Spain**	**14.8**	Austria	25.52	**Bulgaria**	**17.9**	**Spain**	**13.49**	**Iceland**	**20.3**	UK	20.72
**Portugal**	**14.7**	**Hungary**	**24.39**	Romania	17.57	**Portugal**	**12.45**	Norway	19.33	Denmark	20.54
**Hungary**	**14.25**	**Cyprus**	**24.35**	**Croatia**	**17.24**	Romania	12.1	Germany	17.73	Slovenia	19.39
**Bulgaria**	**13.65**	Slovenia	24.26	Luxembourg	14.9	Luxembourg	11.41	Estonia	16.58	**Bulgaria**	**19.28**
**Iceland**	**12.55**	Belgium	23.27	**Hungary**	**14.27**	Belgium	10.69	Ireland	14.2	Estonia	17.36
Ireland	12.45	Luxembourg	22.53	Poland	13.85	Germany	10.03	France	13.69	Poland	17.11
Slovenia	12.45	Romania	21.42	Finland	13.54	Slovakia	9.76	Czechia	13.38	Romania	16.74
Poland	12.4	Ireland	20.87	Austria	12.95	Slovenia	9.46	**Cyprus**	**12.12**	Germany	16.72
Czechia	12.25	France	19.3	**Italy**	**10.68**	Latvia	9.27	Poland	11.52	**Iceland**	**16.45**
Finland	11.7	Slovakia	17.64	Estonia	10.26	Netherlands	8.34	Denmark	10.75	Finland	16.24
UK	11.45	Czechia	16.67	Czechia	10.17	Estonia	8.2	Lithuania	10.75	Ireland	14.22
Germany	11.35	Poland	14.76	Lithuania	9.67	**Croatia**	**8.13**	**Bulgaria**	**9.98**	Norway	13.73
Luxembourg	11.1	**Greece**	**14.7**	**Spain**	**9.47**	**Greece**	**8.12**	Belgium	9.17	**Greece**	**13.66**
Norway	11	**Iceland**	**14.36**	**Portugal**	**9.35**	France	7.32	**Hungary**	**8.95**	Netherlands	13.47
Romania	10.9	Estonia	12.38	France	9.27	Lithuania	7.05	Austria	8.19	Luxembourg	12.95
Slovakia	10.65	**Bulgaria**	**11.57**	Belgium	6.64	Poland	6.4	Luxembourg	7.89	**Italy**	**12.9**
Austria	10.4	Latvia	11.51	Ireland	6.28	Czechia	6.35	Slovakia	7.77	**Croatia**	**11.74**
France	10.3	Netherlands	11.38	Latvia	5.02	Sweden	5.61	**Greece**	7.42	**Cyprus**	**10.39**
Denmark	9.25	Lithuania	9.37	Slovenia	2.99	Finland	4.93	**Croatia**	6.96	**Portugal**	**9.35**
Lithuania	9.1	Finland	5.27	**Iceland**	**2.88**	Ireland	4.52	**Malta**	6.56	Sweden	8.54
Netherlands	9.1	UK	4.61	UK	2.28	**Iceland**	**4.18**	**Spain**	4.85	Czechia	7.16
Latvia	9	Denmark	4.21	Sweden	1.27	Denmark	3.26	**Portugal**	4.73	**Hungary**	**7.03**
Belgium	8.6	Germany	2.56	Norway	0.64	Norway	3.08	Romania	3.79	Austria	6.48
Estonia	8.4	Sweden	1.63	Netherlands	0.39	UK	2.73	Slovenia	3	Slovakia	5.7
Sweden	8.2	Norway	0.04	Denmark	0.21	Austria	0.52	**Italy**	**2.41**	**Malta**	**2.94**

Countries with a higher prevalence of childhood obesity consume more “promoter” and fewer “inhibitor” antibiotics. Glossary: J01A: tetracycline, J01CA: broad-spectrum, beta-lactamase-sensitive penicillin, J01CR: broad-spectrum, beta-lactamase-resistant penicillin, J01D: cephalosporin, J01M: quinolones.

## Data Availability

Not applicable.

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
