# Peer review of "“Growth-Promoting Effect” of Antibiotic Use Could Explain the Global Obesity Pandemic: A European Survey"

_antibiotics, 2022, doi:10.3390/antibiotics11101321_

Round 1
Reviewer 1 Report
The authors discussed the association between obesity and overall antibiotic consumption. To prove their hypothesis, they made statistical analyses using a database of the average yearly antibiotic consumption in childhood and adult obesity prevalence featured Obesity Atlas. In the last decades, highly investigated the human microbiome, which is known as the second immune system. I enjoyed reading this manuscript and found only a few typing errors. I have only one question for the authors, what do they think about the negative correlation between tetracycline and broad-spectrum, beta-lactamase-sensitive penicillin antibiotic consumption and obesity? I believe those highlighted antibiotics have no prevention effect on obesity, only a fact that we use them less. It should be described briefly in the discussion section.
Author Response
Answer to Reviewer 1.
Thank you for your kind opinion as:
The authors discussed the association between obesity and overall antibiotic consumption. To prove their hypothesis, they made statistical analyses using a database of the average yearly antibiotic consumption in childhood and adult obesity prevalence featured Obesity Atlas. In the last decades, highly investigated the human microbiome, which is known as the second immune system. I enjoyed reading this manuscript and found only a few typing errors. I have only one question for the authors, what do they think about the negative correlation between tetracycline and broad-spectrum, beta-lactamase-sensitive penicillin antibiotic consumption and obesity? I believe those highlighted antibiotics have no prevention effect on obesity, only a fact that we use them less. It should be described briefly in the discussion section.
Answer: An explanatory sentence was included in the discussion: Antibiotic classes showing negative associations with obesity figures might inhibit the microbial production of molecules promoting obesity.
Reviewer 2 Report
The current manuscript presents an interesting subject, but all sections should be considered to be revised:
- Throughout the manuscript, a revision of the English used is required, as long sentences are observed in several places.
- To consider this manuscript an article, the authors must reformulate it better and also deal with other notions related with the topic; it looks like a short communication.
- Perhaps the title can be more correct in English in this version: 'Growth-promoting effect' of antibiotic use could explain the global obesity pandemic: a European survey; however, it depends on the authors' desire for improvement.
- The abstract creates confusion as it is formulated; the aim is unclear and appears to be missed. The introduction should be rephrased correctly at the right time. Please improve the last sentence, lines 27-30. A suggestion: sound better using “indicated” instead of “indicates”.
- The introduction section should be revised. Enter the full names of the acronyms first, and then use the respective one throughout the text; examples: in line 38 for the word “WHO” and BMI in line 52. The links on lines 49 and 56 can be expressed as references. A recent study can be helpful to introduce other notion important for the microbiome: [Int J Environ Res Public Health. 2022 Jun 30;19(13):8056. doi: 10.3390/ijerph19138056. PMID: 35805711].
- Section 2 of line 78 is not necessary, it can be canceled and reformulated; as it is presented creates more confusion, because a part of it is a repetition of the introduction. The authors need to point out their aim better and link it clearly with the hypothesis. The long sentence in lines 84-88 isn’t clear.
- The materials and methods section must be organized into subsections, especially create the separate “Statistical Analysis” section in order to be clear with the methodology used. You can't just say “Statistics" and continuing with the description of the tables. Tables 1,2,3 and figures 1,2 are part of the results that cannot be cited in this section, apart from this, the respective table must be inserted where it is mentioned for the first time, for example after line 144 for the tables 1 and 2 and so on for the rest. In line 95, please delete (11 years), not necessary. Also put the full names of ECDC, DID and ATC before their acronyms.
- The results section must also be organized into subsections according to the materials and methods section, explaining in detail and clearly the results obtained and not in general. For each table put a title and the rest should be put as a footnote; there is no sense that huge title for each table. In the legend of each figure, please mention the statistical analysis used.
- The discussion should be reformulated. The second part of the sentence in lines 201-203 is not so clear. On line 205 put the full name before LPS. Authors should discuss following a chronological aspect based on their findings. It is better to add the limitations at the end of the discussion not in the conclusions section.
- The conclusions are not sufficient as presented; the conclusions of this new study need to be demonstrated, seems more part of the discussion. Better to delete and adding the right conclusions. Perhaps the sentence in line 278 can be changed, but without references.
Author Response
Reviewer 2.
Thank you for your remarks/advice. The answers are following your observations:
The current manuscript presents an interesting subject, but all sections should be considered to be revised:
- Throughout the manuscript, a revision of the English used is required, as long sentences are observed in several places.
Answer: The English used had been rechecked and corrections were applied by using the “Grammarly” program. Some long sentences were modified.
- To consider this manuscript an article, the authors must reformulate it better and also deal with other notions related with the topic; it looks like a short communication.
Answer: Corrections were made as indicated. I do not think that a manuscript of 10 pages could be considered a short communication.
- Perhaps the title can be more correct in English in this version: 'Growth-promoting effect' of antibiotic use could explain the global obesity pandemic: a European survey; however, it depends on the authors' desire for improvement.
Answer: The title was modified as proposed.
- The abstract creates confusion as it is formulated; the aim is unclear and appears to be missed. The introduction should be rephrased correctly at the right time. Please improve the last sentence, lines 27-30. A suggestion: sound better using “indicated” instead of “indicates”.
Answer: The aim of the study is indicated: For identifying antibiotic classes which might promote or inhibit obesity-related dysbiosis……………….. Corrections were made.
- The introduction section should be revised. Enter the full names of the acronyms first, and then use the respective one throughout the text; examples: in line 38 for the word “WHO” and BMI in line 52. The links on lines 49 and 56 can be expressed as references. A recent study can be helpful to introduce other notion important for the microbiome: [Int J Environ Res Public Health. 2022 Jun 30;19(13):8056. doi: 10.3390/ijerph19138056. PMID: 35805711].
Answer: The proposed corrections were performed. The suggested article (Odorici A, Colombari B, Bellini P, Meto A, Venturelli I, Blasi E. Novel Options to Counteract Oral Biofilm Formation: In Vitro Evidence. Int J Environ Res Public Health. 2022 Jun 30;19(13):8056. doi: 10.3390/ijerph19138056. PMID: 35805711; PMCID: PMC9265889.) deals with oral biofilm production and its topic do not fit into the scope of our manuscript.
- Section 2 of line 78 is not necessary, it can be canceled and reformulated; as it is presented creates more confusion, because a part of it is a repetition of the introduction. The authors need to highlight their aim and link it clearly with the hypothesis. The long sentence in lines 84-88 isn’t clear.
Answer: The sentence 78 (Antibiotics are known to disrupt microbiota composition and while a rapid recovery is observed following short-term antibiotic treatment, pervasive effects may be obtained after repeated antibiotic perturbations [17, 18}} you mention is not a repetition of anything (I cannot find it) and I actually don’t know what kind of confusion might arise from it, or you might think of a different sentence. The long sentence was corrected and the aim of the investigation was identified as: We aimed to identify antibiotic classes, which might promote or inhibit obesity-related dysbiosis.
- The materials and methods section must be organized into subsections, especially create the separate “Statistical Analysis” section in order to be clear with the methodology used. You can't just say “Statistics" and continuing with the description of the tables. Tables 1,2,3 and figures 1,2 are part of the results that cannot be cited in this section, apart from this, the respective table must be inserted where it is mentioned for the first time, for example after line 144 for the tables 1 and 2 and so on for the rest. In line 95, please delete (11 years), not necessary. Also put the full names of ECDC, DID and ATC before their acronyms.
Answer: The section was organized in different paragraphs without subheadings and the tables and figures were inserted into the “Results” section, but their place might be decided by the layout editor. Other changes had been performed as proposed.
- The results section must also be organized into subsections according to the materials and methods section, explaining in detail and clearly, the results obtained and not in general. For each table put a title and the rest should be put as a footnote; there is no sense that huge title for each table. In the legend of each figure, please mention the statistical analysis used.
Answer: The result section has been organized similarly in paragraphs without subheadings according to the previous section. The results are explained in the discussion section. Tables (title, footnote) are corrected. The description of the statistical analysis is detailed in the previous section and the result of the statistics are featured at the bottom of Table 1 and 2 also.
- The discussion should be reformulated. The second part of the sentence in lines 201-203 is not so clear. On line 205 put the full name before LPS. Authors should discuss following a chronological aspect based on their findings. It is better to add the limitations at the end of the discussion not in the conclusions section.
Answer. The “Discussion” section was modified as requested. The chronology of our analysis is indicated in the section of “Materials and methods” and no “chronological” findings could be featured as it is not a chronological comparison, only a single time frame for antibiotics (2008-18) had been compared to a single obesity database (2019).
- The conclusions are not sufficient as presented; the conclusions of this new study need to be demonstrated, and seem more part of the discussion. Better to delete and add the right conclusions. Perhaps the sentence in line 278 can be changed but without references.
Answer: The “Conclusion” had been replaced
Reviewer 3 Report
This study assessed the association between antibiotic consumption and obesity. Althoug this topic is interesting, many confouding factors cannot be evaluated. In addition, the authors found that strong, positive associations were estimated between childhood obesity and the total consumption of systemic antibiotics, broad-spectrum, beta-lactamase resistant penicillin, cephalosporin, and quinolone, and a negative correlation was found with the consumption of tetracycline, broad-spectrum, beta-lactamase-sensitive penicillin, narrow-spectrum, beta-lactamase sensitive penicillin. However, it is difficult to explain the possible mechanisms about these differences. Thus, I wonder whether these findings could be a co-incidence. More anlaysis is needed.
Author Response
Thank you for your observations and advice.
This study assessed the association between antibiotic consumption and obesity. Although this topic is interesting, many confounding factors cannot be evaluated. In addition, the authors found that strong, positive associations were estimated between childhood obesity and the total consumption of systemic antibiotics, broad-spectrum, beta-lactamase resistant penicillin, cephalosporin, and quinolone, and a negative correlation was found with the consumption of tetracycline, broad-spectrum, beta-lactamase-sensitive penicillin, narrow-spectrum, beta-lactamase sensitive penicillin. However, it is difficult to explain the possible mechanisms about these differences. Thus, I wonder whether these findings could be a co-incidence. More analysis is needed.
Answer: The action of different antibiotics on the gut flora is very different. This is a well-known phenomenon and appropriately documented in the literature. It explains the differences in the effect of different classes of antibiotics. The modified gut flora is “leaking” its products into the circulation or via the neurons, triggering different reactions through the gut-brain axis from the central nervous system. This mechanism had been analyzed in several publications already and many of them are mentioned in the References.
Reviewer 4 Report
The topics is really hot and extremely actual.
I recommend a few modification.
Double check the English.
Please describe the abbreviations when you use them for the first time
The figure 2 is extremally hard to read, maximize the writing.
The conclusion need to clear and specific. My recommendation is to focus on short conclusion.
Thank you again for the opportunity to review this interesting manuscript.
Author Response
Thank you very much for your observations/proposals
The topic is really hot and extremely actual.
I recommend a few modification.
Double check the English.
Please describe the abbreviations when you use them for the first time
The figure 2 is extremely hard to read, maximize the writing.
The conclusion need to clear and specific. My recommendation is to focus on a short conclusion.
Thank you again for the opportunity to review this interesting manuscript.
Answer: The suggested corrections had been performed as requested.
Round 2
Reviewer 2 Report
Thank you to the authors for the corrections made according to the comments made! But there are still few things to recheck before the publication:
- I understand the authors' weariness in carrying out this study, but to call a study an "Article" must be broad and include other fields of medicine and not just a piece of it, because antibiotics are used “en masse” for the oral diseases and so on. For this was the suggestion, it was not said to refer only to that article, but also to see other recent studies in manner to enhance the section. The oral microbiome must be treated better as a concept, on which "other notions" was referred in the last comment.
- Regarding to the English, it refers to a "native person" and not a grammar program. Also, in this version the English leaves something to be desired.
- In several articles on antibiotics, it is preferred to express the links as references and for this was the suggestion.
- The referee does not agree to keep lines 78-83 in section 2, because it is part of the introduction and an aim paragraph contains no references, but clearly describes the aim or hypothesis as in this study.
- The suggestion on the subsections was to give a new “look” to the manuscript; for example, lines 129-133 must be in line 93, so to start the section with this statement. The authors decide whether they like this suggestion.
- The title of table 3 is not suitable, so the frame does not make sense in the title; authors must separate and create the right title, the rest as a footnote.
- It is not clear what the authors have inserted in figures 1 and 2 of the X axes… ??? The legend is outside the figure not inside. So, my last comment was to mention the statistical analysis used for the graphs, and in this case what is the X-axis called?
Author Response
To the Reviewer No. 2.
Thank you very much for your observations and I want to address your remarks as I can. Our manuscript has been reviewed by four (4) Reviewers and 3 of them accepted the modified manuscript, together with the used English. My answers are found under your remarks:
- I understand the authors' weariness in carrying out this study, but to call a study an "Article" must be broad and include other fields of medicine and not just a piece of it, because antibiotics are used “en masse” for the oral diseases and so on. For this was the suggestion, it was not said to refer only to that article, but also to see other recent studies in manner to enhance the section. The oral microbiome must be treated better as a concept, on which "other notions" was referred in the last comment.
General answer: The topic of the association between the altered microbiome (dysbiosis, antibiotics) and the development of different metabolic (obesity, diabetes), neurodegenerative (Parkinson’s, Alzheimer’s, Multiple Sclerosis, etc.) and malignancies (solid tumors, hematological malignancies), plus several other ailments attracted extensive research activity in the last decade and resulted in thousands of publications. This association is well known to those researchers who are active in this area, this is why I did not consider it necessary to describe the related similar fields. I have published over a dozen of such articles in English. The last one can be found in Antibiotics (Ternák G, Németh M, Rozanovic M, Márovics G, Bogár L. Antibiotic Consumption Patterns in European Countries Are Associated with the Prevalence of Parkinson's Disease; the Possible Augmenting Role of the Narrow-Spectrum Penicillin. Antibiotics (Basel). 2022 Aug 23;11(9):1145. doi: 10.3390/antibiotics11091145. PMID: 36139924.) and some more are considered for publications (autism, Down syndrome, etc.) also. Our present manuscript deals with antibiotic consumption and obesity, raising the concept of the similarity between the origin of human obesity and the well-known phenomenon of the effect of antibiotic-enriched fodder and growth promotion in food animals. I did not consider extending the scope of the manuscript much beyond this subject (obesity), but I have referred to it (16) and I have added some explanatory words as: …….and several other ailments {16}.
- Regarding to the English, it refers to a "native person" and not a grammar program. Also, in this version the English leaves something to be desired.
Answer: Some minor corrections have been introduced by a “native person”.
- In several articles on antibiotics, it is preferred to express the links as references and for this was the suggestion.
Answer: Similarly, other articles feature links within the text as well as I did in my previous article published in Antibiotics.
- The referee does not agree to keep lines 78-83 in section 2, because it is part of the introduction and an aim paragraph contains no references, but clearly describes the aim or hypothesis as in this study.
Answer: The lines were moved to the end of the Introduction (blue marking color)
- The suggestion on the subsections was to give a new “look” to the manuscript; for example, lines 129-133 must be in line 93, so to start the section with this statement. The authors decide whether they like this suggestion.
Answer: My previous publications had the same structure, “look”, and I did not want to change it. The lines were moved as requested (blue marking color).
- The title of table 3 is not suitable, so the frame does not make sense in the title; authors must separate and create the right title, the rest as a footnote.
Answer: The title of Table 3. exactly describes the content of this database (Rank order of childhood obesity and the rank order of “promoter” and “inhibitor” antibiotic consumption in 30 European countries). I have no better proposal for it. The frame was erased.
- It is not clear what the authors have inserted in figures 1 and 2 of the X axes… ??? The legend is outside the figure not inside. So, my last comment was to mention the statistical analysis used for the graphs, and in this case what is the X-axis called?
Answer: It was a technical error, corrected.
A few sentences were added to the end of the discussion with a new reference (35). (Green marking color).
Reviewer 3 Report
The authors resposne well, so I have no more suggestions.
Author Response
Thank you for accepting my response.